# QUANTIZATION ENHANCED CROSS-MODAL ALIGNMENT FOR GENE EXPRESSION PREDICTION

## ABSTRACT

In modern healthcare, whole-slide histological images (WSIs) provide information on tissue structure and composition at the microscopic level. Integrating WSIs and gene expression profiles enhances cancer diagnosis and treatment planning, advancing clinical care and research. However, spatial transcriptomics is costly and requires a long sampling time. The intrinsic correlation between histological images and gene expressions offers the potential for predicting spatial transcriptomics using Hematoxylin-Eosin (H&E) stained WSIs to reduce time and resource costs. Although existing methods have achieved impressive results, they ignore the heterogeneity between modalities of image and gene expression. In this paper, we propose a Quantized Cross-modal Alignment (QCA) that exploits cross-modal interactions to address the issue of modal heterogeneity. Considering the interference of gene-unrelated image features, we develop a Gene-related Image Feature Quantizer (GIFQ) to capture the gene-related image features. Meanwhile, we develop an Asymmetric Cross-modal Alignment (ACA) approach, which facilitates the model to generate discriminative predictions from similar visual presentations. In addition, to fix the discriminability reduction, a Discriminability-Enhancing Regularization (DER) is further devised to regularize both the virtual and real gene features. Experimental results on a breast cancer dataset sampled by solid-phase transcriptome capture elucidate that our QCA model achieves state-of-the-art results for accurate prognostication of gene expression profiles, increasing the performance by 13% at least. Our method utilizes deep learning technology to delineate the correlation between morphological features and gene expression, furnishing new perspectives and instruments for disclosing biomarkers in histological conditions. The code will be released.

## 1 INTRODUCTION

In modern healthcare, whole-slide histological images (WSIs) are the gold standard for cancer diagnosis, offering detailed views of tissue samples that enhance pathologists' accuracy in identifying cancer Ghaznavi et al. (2013); Lu et al. (2021); Van der Laak et al. (2021). The combination of WSIs and gene expression profiles allows for visualizing tissue morphology alongside the corresponding gene expression profiles, thereby facilitating a more comprehensive understanding of cancer's molecular mechanisms. However, implementing spatial transcriptomics is a high cost and requires extended sampling periods. This has prompted the exploration of alternative methodologies that can mitigate these drawbacks while still harnessing the valuable information provided by WSIsRao et al. (2021).

The intrinsic correlation between the morphological features of Hematoxylin-Eosin (H&E) stained WSIs and the underlying gene expression patterns presents a promising avenue for developing predictive modelsSchmauch et al. (2020). By leveraging the wealth of data present in H&E stained WSIs, researchers aim to predict spatial transcriptomics data, significantly reducing the time and resource expenditure associated with traditional spatial transcriptomics techniques. Previous studies have demonstrated that sequencing-based spatial transcriptomics data contain thousands of genes, many of which vary insignificantly Williams et al. (2022); Marx (2021). Predicting high-variable genes could be beneficial for disease diagnosis and drug selection. Therefore, we expect to develop models to predict the expressions of these genes based on the H&E stained WSI.

Despite the extensive and profound development of algorithms for deriving information from spatial transcriptomics data Lu et al. (2020); Chen et al. (2021); Shmatko et al. (2022), methodologies for predicting spatial transcriptomics data with histological image information have not yet achieved comparable advancements. Deep learning techniques frequently encounter the interference of gene-unrelated image features and exhibit low performance He et al. (2020); Pang et al. (2021); Zeng et al. (2022). Moreover, a methodology Xie et al. (2024) based on image retrieval tends to yield predictions resembling the predominant samples within the top K selection and necessitates enhanced robustness against outliers and sample imbalance. While specific probability-based models underscore the relative associations of spatial gene expressions, their prediction is always around the mean values of each gene, lacking discriminability Srivastava et al. (2017); Thanh-Tung & Tran (2020). In addition, all the existing methods employ total normalization for gene expression profiles to predict the proportion of gene expressions at sampling points rather than predicting the actual value of gene expression.

Previous methods generally assumed that the conditional probability distribution of gene expression profiles was continuous log-normal Limpert et al. (2001). Actually, one crucial pre-processing step in gene expression prediction is to perform log1p-transformation, which will significantly change the probability distribution of the transformed data. As shown in Figure 1, the log1p-transformation transfers the gene expression profiles from log-normal distribution to normal distribution, leading to the different value densities between high- and low-expressed ranges. This phenomenon reveals the necessity to increase the variation of low-expressed genes, thereby predicting gene expressions with high accuracy. Moreover, the sparse distribution of sampling spots in spatial transcriptomes causes the spatial proximity of image patches to have a relatively small effect on their content similarity, failing to improve prediction performance through spatial information effectively.

To address the above challenges, this paper proposes a novel method termed the Quantized Cross-modal Alignment (QCA) model, which exploits cross-modal interactions between the image and the gene information. Considering the interference of gene-unrelated image features, we develop a Gene-related Image Feature Quantization (GIFQ) to selectively compress image features, highlighting the gene-related features. Meanwhile, we propose an Asymmetric Cross-modal Alignment (ACA), facilitating the model to generate discriminative predictions from similar visual presentations. In addition, a Discriminability-Enhancing Regularization (DER) is devised to regularize both the virtual and real gene features, consequently fixing the discriminability reduction and enabling the prediction of the gene expression profiles. The main contributions of this study can be summarized as follows:

- The proposed QCA method offers a solution for predicting spatial transcriptomics from WSI images through cross-modal interaction techniques. Based on the multimodal translation model, our method incorporates gene expression profiles to effectively align the image and gene models and optimize gene-related image features by utilizing the ACA and GIFQ. The DER is also devised to fix the discriminability reduction.

- The QCA model achieves state-of-the-art prediction results on a breast cancer dataset sampled by solid-phase transcriptome capture, verifying the model's effectiveness in predicting spatial transcriptomics from WSI images. The model significantly surpasses existing methods in terms of predictive correlation for marker genes (MGs), high-variable genes (HVGs), and high-expressed genes (HEGs).

- This study is the first to predict the actual gene expression profiles instead of the expression proportion using deep learning techniques. Our method can uncover genes related to biological histological features, establishing a statistical connection between morphological features and gene expressions.

## 2 RELATED WORKS

### 2.1 TECHNIQUES IN GENE EXPRESSION PREDICTION FROM HISTOLOGICAL IMAGES

Following the advent of ST-Net, a convolutional neural network (CNN) architecture rooted in deep learning He et al. (2020), techniques such as HisToGene Pang et al. (2021) and HistST Zeng et al. (2022) have strived to enhance prediction precision by integrating WSI's spatial information with gene expression proportions. However, multimodal translation models frequently encounter the

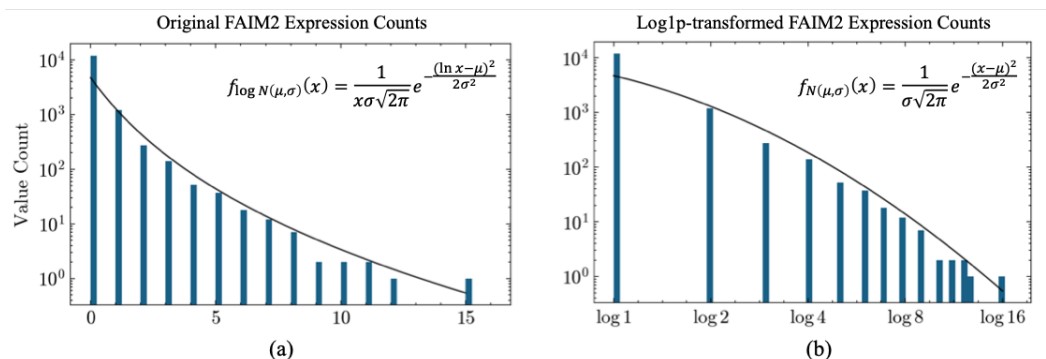

Figure 1: FAIM2 expression counts before and after log1p-transformation (blue bars) with corresponding probability distribution (black curves). (a) Original FAIM2 expression counts fit a log-normal distribution, and the distribution is in integer form. (b) Although log1p-transformed FAIM2 expression counts follow a normal distribution, they present different value densities between high- and low-expressed ranges.

interference of gene-unrelated image features, thereby failing to extract the biological heterogeneity encapsulated in H&E stained images effectively. The BLEEP Xie et al. (2024), grounded in an image lookup methodology, performs prediction by identifying and assigning weights to the nearest neighbors among stored reference sample features. Nonetheless, this method necessitates enhanced robustness against sample imbalance and tends to predict outcomes resembling the predominant samples within the top K selection. Approaches like Xfuse Bergenstråhle et al. (2022); Zhang et al. (2024) produce super-resolution gene expression profiles by fusing WSI with reference spatial transcriptomics data. Although these probability-based models highlight the relative associations of spatial gene expressions, their predictive outcomes often revolve around the mean values of each gene, lacking discriminability.

While specific methodologies have endeavored to tackle the prediction challenges associated with spatial transcriptomics data, they still need to overcome limitations in circumventing the heterogeneity between modalities and augmenting prediction accuracy. Consequently, it is imperative to investigate novel methodologies to address these challenges and bolster models' predictive capabilities in the context of spatial transcriptomics data.

## 2.2 FINITE SCALAR QUANTIZATION

Vector quantization technology Gray (1984) has found extensive application in deep learning, particularly in data compression Barnes et al. (1996). This technology achieves effective feature compression by mapping continuous latent spaces to discrete coding spaces. In the VQ-VAE model Van Den Oord et al. (2017), input data is first mapped to the latent space through an encoder network, subsequently assigned to the nearest codebook vector via a quantization process, and finally reconstructed or generated to resemble the input data through a decoder network. Constructing a discrete latent space enhances the model stability and the convergence of the training process.

Finite Scalar Quantization (FSQ) technology Mentzer et al. (2023) is a technique used to optimize vector quantization and is particularly applicable in domains such as image generation. The primary advantage of FSQ lies in its ability to effectively mitigate the codebook collapse issue prevalent in VQ-VAE. Moreover, FSQ is characterized by its simplicity of implementation, as it obtains an implicit codebook vector by projecting VQ-VAE representations into a lower-dimensional space and quantizing each dimension. This approach has demonstrated competitive performance in feature compression and possesses fewer parameters than VQ-VAE.

Inspired by FSQ, this paper proposes a GIFQ, which aims to compress image features and selectively preserve image features associated with gene expressions through learning codebook vectors. This approach seeks to enhance image features in deep learning models by reducing the interference of gene-unrelated features.

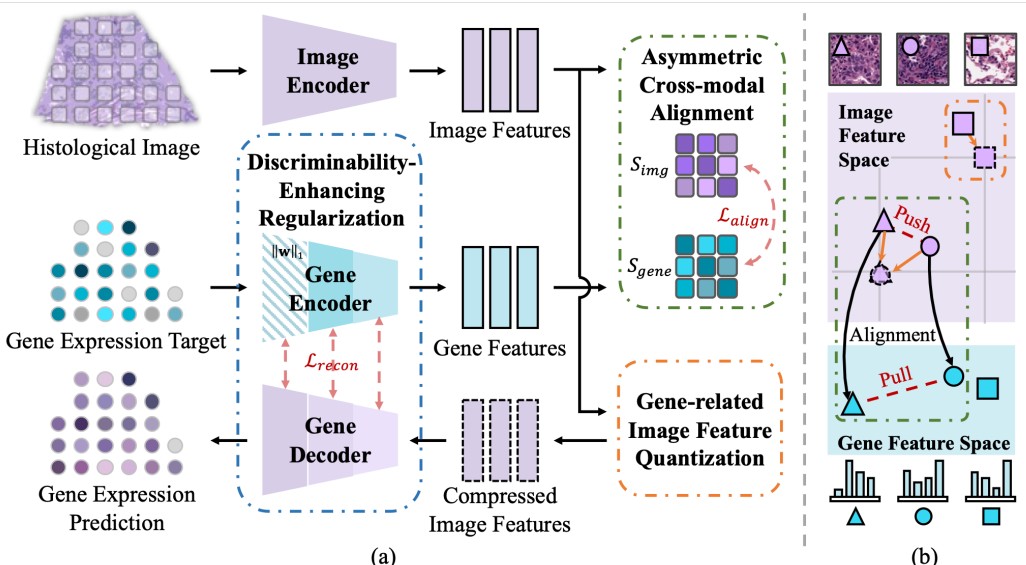

Figure 2: (a) Overall framework of QCA. QCA employs an ACA to align image features with expression features and calculate the alignment loss. Simultaneously, the image features are fed into a GIFQ to obtain compressed image features. Next, the features are regularized by a DER to calculate the reconstruction loss. Finally, the features are sent to the gene expression decoder to generate the prediction. (b) Operations of ACA and GIFQ in Feature Spaces. The ACA could push image features away and pull gene expression features closer by constraining the similarity matrices $S_{img}$ and $S_{gene}$. GIFQ achieves feature compression by mapping image features to their nearest integer space (grey grid).

## 3 METHODS

Existing methodologies for generating gene expression profiles from histological images encounter two major challenges: (1) failing to generate discriminative predictions from similar visual presentations and (2) being sensitive to the interference of gene-unrelated image features. As depicted in Figure 2, we address these challenges through an Asymmetric Cross-modal Alignment (ACA) and a Gene-related Image Feature Quantization (GIFQ). Due to the reduction in discriminability between gene expression features and image features by the ACA and GIFQ, a Discriminability-Enhancing Regularization (DER) is devised to enhance the discriminability of these features through regularization, thereby achieving precise gene expression prediction. The details are provided in Algorithm 1, where $B$, $W$, $C$ represent batchsize, image width, and number of gene categories.

### 3.1 ASYMMETRIC CROSS-MODAL ALIGNMENT

In gene expression prediction tasks, neither probability-based models nor multimodal translation models have failed to generate discriminative predictions from similar visual presentations. However, samples with similar visual presentations may exhibit different gene expression features in reality. Aligning gene expression features with image features can emphasize the correlation between the modalities, effectively addressing the issue.

One straightforward way is to constrain the features of the modalities to be identical symmetrically Radford et al. (2021). However, it fails to highlight the similarity in the intra-modality. To address this problem, we propose an Asymmetric Cross-modal Alignment (ACA) that calculates the similarity matrices between image features and gene expression features separately and uses the cross-entropy loss function to align these matrices, thereby guiding the image feature extractor to learn features related to gene expression. Specifically, in scenarios where "image features are dissimilar, but gene expressions are similar," the confusion has only a minimal impact. In contrast, scenarios where "gene expressions are dissimilar, but image features are similar" require attention. For similarity

**Algorithm 1** Quantized Cross-modal Alignment Training Algorithm

---

**Input:** H&E image patches ($I \in \mathbb{N}^{B \times 3 \times W \times W}$), log1p-transformed gene expression profiles ($X \in \mathbb{N}^{B \times C}$), pre-trained image encoder ($g_{encoder}$), gene encoder ($f_{encoder}$), gene decoder ($f_{decoder}$), gene-related image feature quantizer ($Q$)

**Output:** Virtual gene expression profiles ($\hat{X} \in \mathbb{N}^{B \times C}$)

1: Initialize variables
2: $H_{img} \leftarrow g_{encoder}(I)$          ▷ Extract image features
3: $H_{encoder}^{(0)} = X$
4: **for** $l \leftarrow 1$ **to** $L$ **do**
5:      $H_{encoder}^{(l)} \leftarrow f_{encoder}(H_{encoder}^{(l-1)})$
6:      $\mathbf{w}^{(l)} \leftarrow weight(f_{encoder}^{(l)})$
7: **end for**
8: $H_{gene} = H_{encoder}^{(L)}$          ▷ Extract gene expression features
9: $S_{img}, S_{gene} \leftarrow (H_{img}^T \cdot H_{img}), (H_{gene}^T \cdot H_{gene})$
10: $S_{min}, S_{max} \leftarrow softmax(\min(S_{img}, S_{gene})), softmax(\max(S_{img}, S_{gene}))$
11: $\mathcal{L}_{align} \leftarrow (1 - \alpha) \cdot CE(S_{img}, S_{min}) + \alpha \cdot CE(S_{gene}, S_{max})$
         ▷ Asymmetric Cross-modal Alignment
12: $H_{quantized}, H_{codebook} \leftarrow Q(H_{img})$      ▷ Gene-related Image Feature Quantization
13: $H_{decoder}^{(L)} = H_{quantized}$
14: **for** $l \leftarrow L$ **to** $1$ **do**
15:      $H_{decoder}^{(l-1)} \leftarrow f_{decoder}(H_{decoder}^{(l)})$
16: **end for**
17: $\mathcal{L}_{recon} \leftarrow \sum_l MSE(H_{encoder}, H_{decoder}) + \beta \cdot \sum_{i=1}^{C} ||\mathbf{w}_i^{(1)}||_1$
         ▷ Discriminability-Enhancing Regularization
18: $\mathcal{L} = \mathcal{L}_{align} + \mathcal{L}_{recon}$
19: $\hat{X} \leftarrow f_{decoder}(H_{decoder}^{(0)})$
20: **Return** $\mathcal{L}, \hat{X}, H_{codebook}$

---

matrices from different modalities, $S_{img}, S_{gene}$, the model focuses on the latter scenario through an asymmetric loss function:

$$\mathcal{L}_{align} = (1 - \alpha) \cdot CE(S_{img}, \min(S_{img}, S_{gene})) + \alpha \cdot CE(S_{img}, \max(S_{img}, S_{gene})), \quad (1)$$

where $\mathcal{L}_{align}$ represents the alignment loss, $CE(\cdot)$ represents the cross-entropy loss and $\alpha$ is a hyperparameter that regulates the fitting rate of the two modality feature extractors due to the varying proportions of information related to the other modality.

Furthermore, the asymmetric feature alignment has increased the discriminability of virtual gene expression features while simultaneously reducing the discriminability of real gene expression features. The issue will be addressed by the DER subsequently.

## 3.2 GENE-RELATED IMAGE FEATURE QUANTIZATION

The interference of gene-unrelated image features significantly impacts gene expression prediction with histological images, leading to the issue of mode collapse, i.e., generating prediction around the mean of each gene. Hence, the model employs a Gene-related Image Feature Quantization (GIFQ) to compress the image features. This module preserves gene-related image features through a learnable discrete codebook $c_k$ quantizing the encoding features.

$$z_q(x) = \arg\min_{c_k} ||z_e(x) - c_k||_2, \quad (2)$$

where $z_e(x)$ and $z_q(x)$ represent the image feature before and after quantization, respectively, and $c_k$ represents the codebook. To selectively preserve image features within the codebook vectors,

the vector quantization module utilizes the Straight-Through Estimator (STE) method Bengio et al. (2013) to facilitate the passage of gradients during the training phase. This approach solves the issue that the gradients can not be transmitted during the computation of $z_q(x)$.

$$\nabla z_e(x) = \nabla z_q(x). \tag{3}$$

For finite scalar quantization, set $\{c_k\} = \{\lfloor z_e(x) \rfloor \mid x \in \mathbb{R}^n\}$ to fix the issue of low utilization rate when building the codebook vectors through gradient descent. In practical application, we employ the function $round_{STE}(\cdot)$:

$$round_{STE}(x) = x + stopgrad[round(x) - x], \tag{4}$$

where $stopgrad[\cdot]$ represents the cessation of gradient descent, ensuring that the gradient of the function's output is identical to that of $x$. Compression of image features is achieved by constraining the range of the codebook vectors, allowing the model to avoid local minimal around the mean of each gene caused by gene-unrelated image features.

### 3.3 DISCRIMINABILITY-ENHANCING REGULARIZATION

After the feature compression, constraining the range of the codebook vectors may impact the discriminability of the image features. Moreover, while the ACA aligns the image features with gene features and improves the discriminability of the virtual gene features, it concurrently mitigates the discriminability of real gene features by reducing feature differences. Owing to the limited spatial and temporal extent of spatial transcriptomic sampling, high-expressed genes vary greatly compared with low-expressed ones (as shown in Figure 1), leading to the same problem. The model's emphasis on high-expressed genes could improve the issue of low discriminability in gene features.

In our model, a multi-component L1 regularization is deployed to augment the model's awareness of high-expressed genes, thereby enhancing the discriminability of gene expression features. Using the discriminative gene expression features, the discriminability of the decoding image features can be improved, consequently facilitating better prediction of gene expression. Thus, we designed a multi-level feature alignment loss function assuming that the semantic feature levels in each layer are similar to transform image features into virtual gene expression profiles Ronneberger et al. (2015); Cai et al. (2024).

$$L_{recon} = \sum_{i=0}^{C} ||\mathbf{w_i}||_1 + \sum_{l} MSE(H_{encoder}^{(l)}, H_{decoder}^{(l)}), \tag{5}$$

where $L_{recon}$ represents the reconstruction loss, $\mathbf{w}$ represents the weight of the gene encoder's input layer, and $MSE(\cdot)$ represents mean square error.

## 4 EXPERIMENTS

### 4.1 DATA AND EVALUATION METRICS

The data utilized for training are derived from breast cancer datasets sampled by solid-phase transcriptome capture technology. The dataset contains 36 spatial transcriptomes and corresponding frozen sections from 8 patients Andersson (2021); Andersson et al. (2020), which can be accessed via the following link: `https://zenodo.org/records/3957257#.Y4LB-rLMIfg`. The version of the dataset used in this paper is v3.0, distributed under the Creative Commons Attribution 4.0 International (CC BY 4.0) license. We select $224 \times 224$ pixel-sized stained image slices centered on the target sampling spots and extract the spot areas' image features through CTransPath Wang et al. (2022). We selected the top 1000 variable genes for prediction.

Our study primarily employs the Pearson correlation coefficient (PCC) to evaluate the similarity between predicted gene expression profiles and the ground truth. Following the BLEEP Xie et al.

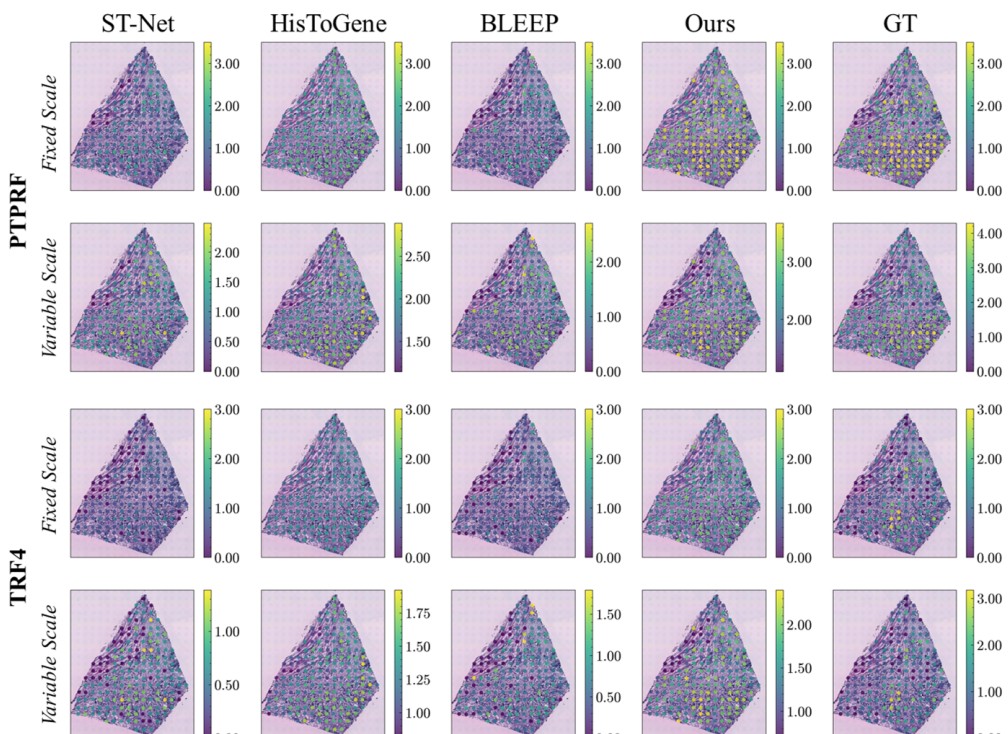

Figure 3: Original and predicted spatially resolved expressions for PTPRF and TRAF4 across various methods, visualized utilizing fixed and variable color scales.

(2024), we focus on the average PCC of the marker genes (MGs), the top 50 high-variable genes (HVGs), and the top 50 high-expressed genes (HEGs) to assess the experimental outcomes. In the ablation experiments, we use mean squared error (MSE) to gauge the model's fitting accuracy.

## 4.2 IMPLEMENTATION DETAILS

Since gene expression profiles exhibit a log-normal distribution, the gene expression profiles utilized in the experiments underwent log1p-transformation processing. Gene expression profiles are in integer form; however, the models compared with the proposed method are used to predict the proportion of gene expression in the same sampling spot. Therefore, we employ a rounded exponential function for gene expression prediction ($\hat{X}$) to accurately reconstruct the predicted results in each model: $\hat{X} \leftarrow \log round_{STE}(\exp \hat{X})$.

The experiments were implemented on the PyTorch 1.10 platform with an Nvidia-GeForce 3090 GPU. For detailed information regarding the experimental setup and implementation, please consult the Supplementary Material.

Table 1: Comparative experiments of different models

| Method | Model Size | Average Correlation | | |
|---|---|---|---|---|
| | | MG | HVG | HEG |
| ST-Net | 78 M | 0.133±0.025 | 0.137±0.034 | 0.299±0.028 |
| HisToGene | 890 M | 0.138±0.002 | 0.151±0.013 | 0.266±0.002 |
| BLEEP | 150 M | 0.138±0.025 | 0.128±0.031 | 0.253±0.040 |
| QCA | 133 M | **0.168**±0.003 | **0.207**±0.015 | **0.340**±0.025 |

Table 2: Ablation experiments on module changes

| Modules | | MSE | Average Correlation | | |
| Alignment | Quantization | | MG | HVG | HEG |
| --- | --- | --- | --- | --- | --- |
| none | ✓ | 0.481±0.010 | 0.162±0.009 | 0.124±0.015 | 0.323±0.023 |
| identical | ✓ | 0.465±0.012 | **0.180**±0.014 | 0.152±0.013 | 0.326±0.018 |
| asymmetric | ✗ | 0.486±0.018 | 0.164±0.010 | 0.137±0.015 | 0.320±0.023 |
| asymmetric | ✓ | **0.454**±0.010 | 0.168±0.003 | **0.207**±0.015 | **0.340**±0.025 |

## 4.3 COMPARATIVE EXPERIMENTS

Table 1 presents the Pearson correlation coefficients between the predicted gene expressions by four methods and the actual gene expression profiles. The observations indicate that the QCA model is not superior in model size. However, for the MGs, HVGs, and HEGs, our QCA model achieves significant performance improvements in predictions, increasing the performance by 21%, 37%, and 13%, respectively.

Recent multimodal translation models frequently confront the challenge of mode collapse in endeavors to synthesize virtual spatial transcriptomics from histological images, as illustrated in Figure 3. Although these deep learning architectures can describe the spatial correlations of gene expression profiles, they often converge to local minima near the mean of each gene. However, the incorporation of the proposed GIFQ addresses this issue. Figure 4 (a) illustrates the t-SNE visualization Linderman et al. (2017) of the latent space embeddings constructed by the QCA model and the corresponding gene expression. This demonstrates that the GIFQ compresses features through feature selection, retaining meaningful features for judging gene expression profiles.

This advancement is crucial for comprehending tumor heterogeneity, pinpointing novel therapeutic targets, and devising personalized treatment strategies. Moreover, the model's facilitation in constructing information regarding biological heterogeneity enhances the precision of simulating intricate biological processes within the tumor microenvironment, offering a robust instrument for investigating tumor development and progression.

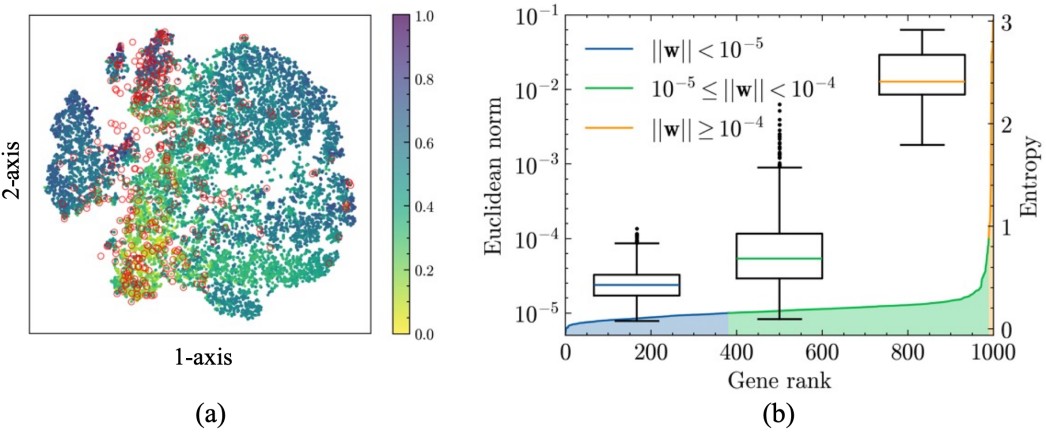

(a)          (b)

Figure 4: (a) t-SNE visualization of codebook vectors and image features by red circles and dots, respectively. The color of the image feature indicates the distance from the neighboring codebook vector. (b) The entropy differences stratified by regularized weights of the gene encoder's input layer. Where the curve represents the input layer of the gene encoder's weight, and the phase line represents the entropy of the genes with high, medium, and low weights

## 4.4 ABLATION STUDY

The ablation experiments presented in Table 2 reveal that the model utilizes an asymmetric cross-modal interaction framework to more accurately predict gene expression profiles using image information, showcasing enhanced precision and stability compared to the computational approach of directly aligning the features of the modalities. In addition, we can observe that the ACA solves the problem of confusing predictions from similar visual presentations through an emphasis on scenarios where "gene expressions are dissimilar, but image features are similar." This can be demonstrated in high-variable genes (HVGs), which are significantly affected by similar predictions and obtain 0.055 improvements compared to the model, which constrains the features to be identical (0.152). In addition, we can also find that identical feature alignment presents a significant improvement over the model without alignment in HVGs.

Regarding selecting the hyperparameter $\alpha$, it is challenging to simultaneously achieve optimal results for all evaluation metrics under specified values of $\alpha$. Consequently, we employ the MSE to attain the most balanced result. As shown in Table 3, the choice of $\alpha = 0.1$ yielded the minimal MSE, suggesting that imposing a relatively higher loss on the image feature extractor is appropriate. This indicates that the model requires more substantial adjustments to image features than regulating gene expression feature extractors.

Besides, we utilized Shannon entropy Lin (1991) to evaluate the discriminability of each gene, with higher entropy indicating greater discriminability in gene expression. Figure 4 (b) depicts the entropy differences stratified by regularized weights. The figure shows that the model can adapt varying amounts of information for different genes. This adjustment has enhanced the model's ability to distinguish gene expression features, particularly those with high entropy. Notably, although the model has achieved relatively stable results in predicting MGs, the average correlation coefficient did not surpass the outcome when $\beta$ was set to 0. This discrepancy might be attributed to the lack of discriminability in MGs compared to HVGs and HEGs, necessitating further analysis of the biological significance of marker genes to enhance the stability of their prediction.

Table 3: Ablation experiments on hyperparameter

| Hyperparameters | | MSE | Average Correlation | | |
|---|---|---|---|---|---|
| $\alpha$ | $\beta$ | | MG | HVG | HEG |
| 0.10 | 0 | 0.472±0.009 | **0.172**±0.022 | 0.150±0.023 | 0.314±0.024 |
| 0.50 | 0.01 | 0.458±0.027 | 0.152±0.017 | 0.179±0.028 | **0.344**±0.034 |
| 0.90 | 0.01 | 0.474±0.015 | 0.166±0.013 | 0.137±0.017 | 0.312±0.011 |
| 0.10 | 0.01 | **0.454**±0.010 | 0.168±0.003 | **0.207**±0.015 | 0.340±0.025 |

## 5 DISCUSSION AND CONCLUSION

In this paper, we propose a Quantized Cross-modal Alignment (QCA) model consisting of three modules. The Asymmetric Cross-modal Alignment (ACA) improves the image features to address the issue of predicting confusing gene expression from image features. The Gene-related Image Feature Quantization (GIFQ) tackles the issue of mode collapse, which frequently occurs in gene prediction. Furthermore, the Discriminability-Enhancing Regularization (DER) enhances the discriminability of both virtual and real gene features to achieve an accurate prediction of gene expressions. The comparative and ablation experimental results indicate that the QCA model is significantly advantageous in predicting gene expression profiles.

However, due to non-biological experimental factors, gene expression from different samples exhibits significant variations, which may limit the absolute performance of the model in generating gene predictions. Advancements in sampling techniques and matched spatial transcriptomic data availability can enable a deeper exploration of the multimodal interaction latent space. At the same time, bioinformatics analysis of gene ontology can help better determine and predict logically related genes.

In summary, our QCA is a deep learning framework that utilizes cross-modal interaction technology to address the adverse properties of gene expression profiles when interacting with other modalities. Our study is the first to predict the actual gene expression profiles using deep learning techniques. It provides new perspectives and tools for a deeper understanding of the complicated relationship between gene expression and morphological structure.

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

## A  APPENDIX

You may include other additional sections here.

