# OpenReview forum: "Quantization Enhanced Cross-modal Alignment for Gene Expression Prediction"
_ICLR.cc/2025/Conference — Submitted to ICLR 2025_

### Official Review · Reviewer_ANtH · 2024-10-30

**Soundness:** 2
**Presentation:** 2
**Contribution:** 2
**Rating:** 5
**Confidence:** 3

**Summary:**

Motivation: Spatial transcriptomics is costly and time-consuming to do. Can spatial gene expression be predicted using whole slide images (WSI)?
Technical contribution: A cross-modal alignment recipe is proposed to find key visual images in predicting gene expression
Strengths: Comparison with strong and recent baselines.
Weakness: Lacking interpretability and training and implementation details.

Justification of score:
Given the missing training details and lack of clarity on some implementation choices, I am recommending a score of 5. I would be happy to reconsider my score after looking at the authors' comments.

**Strengths:**

- While the problem of predicting gene expression from images is not a novel idea, the proposed method has strong gains over the baselines, indicating that their method is able to capture meaningful signals.
- The writing is generally clear to a broad audience. The problem is well formulated and motivated.
- The proposed method has been compared to recent baselines, such as BLEEP.

**Weaknesses:**

**Additional interpretability**
- The authors claim in their abstract that their method can "delineate the correlation between morphological features and gene expression."  This is an interesting application of the proposed method. However, I do not see any interpretability analysis in the paper to support the claim made by the authors (only qualitative reasoning can be done with Figure 3). Can the authors elucidate any correlations between morphological features and gene expression that their method allows to find?

**Validation on additional cohorts**
- I believe that the robustness and generalizability of the proposed method can be tested further as right now it is only limited to one organ and dataset. I urge the authors to validate their method on other organs and datasets (such as kidney or prostate) from the recently published dataset HEST-1K [1].

**Clarifications on WSI patch encoder**
- The authors use CTransPath WSI patch encoder in their study (line 321). However, 10+ foundation models for WSI have been published recently, which have been shown to perform significantly better than CTransPath. Can the authors comment on this choice and try to use a stronger foundation model, such as Virchow [2]?

**Training details**
- It is unclear how the 36 samples from 8 patients where used for training, validation and testing. Can the authors comment on this aspect? Has there been any use of batch correction methods for omics data as spatial transcriptomics data is bound to have strong batch effects.

---
References:
- [1] Jaume, Guillaume, et al. "Hest-1k: A dataset for spatial transcriptomics and histology image analysis." arXiv preprint arXiv:2406.16192 (2024).
APA
- [2] Vorontsov, Eugene, et al. "A foundation model for clinical-grade computational pathology and rare cancers detection." Nature medicine (2024): 1-12.
APA

**Questions:**

- Line 29 of abstract- prognostication is not the right word to be used here as you are not predicting survival/ outcome
- Missing reference: Hest-1k (Jaume, Guillaume, et al. "Hest-1k: A dataset for spatial transcriptomics and histology image analysis." arXiv preprint arXiv:2406.16192 (2024).)
- It seems intuitively that the proposed method requires that the spatial transcriptomics profile and WSI images need to be registered. Can the authors comment on this? If any special registration technique is used, can the authors describe it?
- When presenting measures of predicting a continuous values, it is a good idea to present Pearson and Spearman correlations as they operate under different assumptions. Can the authors comment on why only Pearson correlation was presented and can they validate their results using additional metrics?
- It is unclear what type of gene encoders the authors have used in the study. Can the authors comment on this?

---

### Official Review · Reviewer_tGur · 2024-10-31

**Soundness:** 1
**Presentation:** 2
**Contribution:** 2
**Rating:** 3
**Confidence:** 5

**Summary:**

This paper introduces a deep learning framework called Quantized Cross-modal Alignment for predicting gene expression profiles from H&E images. The framework consists of three components: Asymmetric Cross-modal Alignment (ACA), Gene-related Image Feature Quantization (GIFQ), and Discriminability-Enhancing Regularization (DER). The method shows improved performance on a breast cancer dataset.

**Strengths:**

1. To the best of the reviewer's knowledge, this is the first work using vector quantizations for learning gene features.
1. The method shows performance improvements over other methods.

**Weaknesses:**

1. The asymmetric cross-modal alignment loss lacks a proper theoretical foundation and biological justification.  The authors focus on cases where "gene expressions are dissimilar, but image features are similar" (line 215). but this premise is questionable both in terms of biological relevance and technical feasibility. From a biological perspective, image features can only effectively predict genes directly influencing morphology; genes with minimal morphological impact are inherently difficult to predict from images alone. Furthermore, the ablation studies on marker genes (which are indicators of cellular identity/function) indicate that the asymmetric loss actually degrades model performance. Additionally, I think the asymmetrical loss is formulated differently in pseudo-code and in equation (line 231 vs. line 249), the softmax term is missing.

2. While the authors repeatedly claim their method's superior discriminative power for gene expression prediction, they fail to provide compelling evidence that the model avoids mean-seeking behavior in its predictions. The presentation of gene entropies stratified by encoder weights in Figure 4(b) is tangential and insufficient to support this argument.

3. Discriminability-enhanced regularization is fundamentally flawed.
   1. The approach contradicts biology. Low-expression genes, particularly transcription factors, often play crucial regulatory roles in cellular processes. The decision to emphasize highly expressed genes while penalizing low-expression genes is not biologically grounded.
   2. The authors make claims that the method (DER) is effective for improving predictive variance, but no evidence is provided, where is the ablation for DER?
   3. Line 294 referred to figure 1 to show variance differences of highly- vs. lowly-expressed genes. But Figure 1 is not.
4. The paper lacks a comprehensive discussion of the model architecture, which is essential for understanding and reproducing the work.
5. I suggest the authors add brackets to the in-line references for readability.

**Questions:**

1. How did the authors derive the marker genes? From what literature and how many of them are used in evaluation?

---

### Official Review · Reviewer_PLY6 · 2024-11-01

**Soundness:** 1
**Presentation:** 1
**Contribution:** 2
**Rating:** 1
**Confidence:** 4

**Summary:**

This paper presents a novel Quantized Cross-modal Alignment (QCA) framework for predicting gene expression profiles from H&E stained whole-slide images (WSIs). The method addresses two key challenges: 1) generating discriminative predictions from similar visual presentations, and 2) reducing interference from gene-unrelated image features.

The paper's primary technical contribution is a three-component QCA framework combining asymmetric cross-modal alignment, feature quantization, and discriminability enhancement. The Asymmetric Cross-modal Alignment (ACA) component introduces an innovative approach to handle cross-modal relationships between image and gene expression features. The Gene-related Image Feature Quantization (GIFQ) effectively addresses the mode collapse problem common in previous approaches. The Discriminability-Enhancing Regularization (DER) maintains feature discriminability while managing the tradeoffs introduced by the other components. Although experiments are conducted on a small dataset, a comprehensive ablation study is included in the manuscript to demonstrate the contribution of each component in accurate gene expression prediction. The framework is reported to achieve at least 13% improvement over existing methods for MG, HVG, and HEG predictions.

**Strengths:**

- The background section is well written with good coverage of the previous work and a comprehensive list of references
- The paper is organised well into various sections required for the ease of readability.
- Since spatial transcriptomics is very expensive, accurate generation of gene expression profiles with representation learning methods, such as the proposed approach, could be a cost-effective alternative.
- Comprehensive ablation studies to validate each component.

**Weaknesses:**

- Limited dataset scope (breast cancer only) and sample size (only 8 patients with 36 transcriptomes)
- Dataset needs to be explained in more detail, specifically how accurate is the spot selection with CTransPath method
- Lack of biological validation of predictions
- Limited discussion of computational requirements and clinical integration
- Figure 1 and Algorithm 1 should have dedicated paragraphs to explain the proposed approach. Current Figure 1 is too overwhelming and could be simplified.

**Questions:**

- The definition of "gene expression" is unclear throughout the paper - specifically, whether it refers to whole transcriptome sequencing, DNA microarrays, single cell RNA sequencing, or another method.
- Key terms such as "spatial transcriptomics" and "heterogeneity between modalities" should be introduced and defined, particularly for machine learning audiences unfamiliar with these concepts. Additionally, the meaning of "augmenting prediction accuracy" (line 139, page 2) requires clarification.
- For point # 1 in line 193, why do we need distinct predictions from similar visual representations? Similar morphology and nuclear appearance should generate similar gene expressions if there is a phenotype-genotype correspondence, which is the foundation of this work. If above is incorrect, then how should we ensure that the generated gene expression profiles have any correspondence with the underlying phenotype as then there would be one-to-many phenotype-to-genotype mappings possible?
- For point # 2 in line 194, the reviewer is not sure of what the authors mean by "gene-unrelated" image features - is there a formal definition of gene-related and gene-unrelated image features?
- What is the rationale for the statement made in line 206 "samples with similar visual presentations may exhibit different gene expression features in reality." Is this an assumption or a biologically verified fact?
- Rationale missing for statements made in lines 214-215. Why there is many-to-one correspondence in phenotype-genotype relationship?
- How to appropriately initialise the gene-related image feature quantizer (Q) in Algorithm 1? Are there specific constraints or requirements on it based on the type of gene expression generated?
- How to select hyperparameters alpha and beta - only through experiments?
- The authors only focus on 1000 variable genes for prediction in this manuscript - how does the method scales for several thousand genes profiled through whole transcriptome sequencing? What are the possible limitations of the proposed approach in this case?
- Is the proposed approach directly applicable to the data generated by different types of gene expression technologies (see point 1 above) or what modifications would be required?
- Why the authors have not used the publicly available data from the Xie et. al.'s paper for a fair comparison with the BLEEP method? That data is available here - https://www.ncbi.nlm.nih.gov/geo/query/acc.cgi?acc=GSE240429
- What is the computationally complexity of the proposed approach?
- What is the cost of generation of an erroneous gene expression profiles in general and through the proposed approach? Can the cost might be measured in terms of patient survival or treatment outcomes or some other measure that captures the impact of the proposed approach on downstream clinically important task?

---

### Official Review · Reviewer_zr77 · 2024-11-04

**Soundness:** 2
**Presentation:** 1
**Contribution:** 2
**Rating:** 3
**Confidence:** 3

**Summary:**

This paper attempts to predict spatial gene expression from tissue morphology using two key innovations: 1) developing gene-related image features to filter non-gene-related noise from the image for this prediction task and 2) an asymmetric cross-modal alignment approach.

I generally like the manuscript, but the inconsistencies and open questions outlined below led me to the indicated score. I am willing to increase my score if my concerns are addressed and questions clarified.

**Strengths:**

- The paper is well motivated and more accurate ST prediction would be a highly valuable contribution to oncology and computational pathology
- The linear probing results of the model are significant and especially impressive when accounting for model size
- The outlined distribution shift following the log1p transformation is relevant to most ST and single-cell pipelines.
- The comparison of the ST prediction compared to ground truth is compelling

**Weaknesses:**

- Introduction:
    - Some papers are cited in the wrong places. For example, in L55, you cite Lu et al. (2020), and Chen et al. (2021) in the context of algorithms deriving information from ST data. However, both papers work with bulk sequencing data (TCGA) and do not consider spatial information.
    - L64: It’s unclear what is meant by “total normalization”
    - L73-75: this only holds for older ST technologies. Visium HD or Xenium slides would not suffer this problem, which is worth mentioning
- While Figure 1 presents an interesting insight, it is unclear how this is directly linked to the motivation of the methods. This aspect is heavily emphasised in the first part of the manuscript but not mentioned in the discussion, which makes these sections relatively disjoint.
- Presentation: Some in-line citations (L44, L48) don’t render correctly and all citations don’t follow the ICLR template. Generally, there seems to be something wrong with the LaTeX template of the manuscript, which does not render correctly.
- The code is not provided in the supplementary materials, although strongly encouraged.

**Questions:**

- It is unclear how the distribution shift after the log1p transform motivates the quantized alignment based on the introduction. Could you expand on this part of the motivation?
- How can you be sure that GIFQ actually reduces the interference of gene-unrelated features?
- Training/evaluation: L321 sates that you select top 1000 variable genes for prediction but you later state that you look at top 50 HVG, HEG, and MGs. I assume that you use 1000 variable genes for training and the rest just for evaluation? Are the HVG/HEGs calculated based on the test or train set?
- Method: Can you explain why you opted for vector quantization over other lossy compression methods or latent variable models?
- The ablation study in Table 3 seems inconclusive. Can you explain what you want to show with this and how the fitting rate from Eq. 1 should be chosen?

---

### Meta-Review · Area_Chair_AUDz · 2024-12-16

**Metareview:**

This work introduces a cross-modal alignment framework for predicting spatial transcriptomics from H&E-stained slides via several modules, including Quantized Cross-modal Alignment (QCA), Gene-related Image Feature Quantizer (GIFQ), Asymmetric Cross-modal Alignment (ACA), and Discriminability-Enhancing Regularization (DER). Experimental results show promising improvement over compared SOTA methods.

This paper received 2x reject, 1x strong reject, and 1x marginally below the acceptance threshold ratings from reviewers. The major concerns regarding this paper raised by reviewers centered around the limited clarity in paper writing, method design, and experimental analysis. The reviewers agreed that further clarifications are needed to better support the statement made in this paper. However, the authors did not respond in the discussion phase and the concerns remain unaddressed.

Given the consensus of reviewers, rejection is recommended.

**Additional Comments On Reviewer Discussion:**

Reviewers raised questions challenging the paper writing (incorrect reference format, lack of interpretation of key terms, latex template error, etc.), clarity of methodology and experimental settings, and comprehensiveness of conducted experiments. Yet the authors did not respond during the rebuttal phase, and the quality of the manuscript is not improved.

---

### Decision · Program_Chairs · 2025-01-22

Reject